# The Dynamic Impact of Physical Education Teacher Support on College Students’ Adherence to Exercise: A Cross-Lagged Study from the Perspective of Self-Determination Theory

**DOI:** 10.3390/bs15060802

**Published:** 2025-06-11

**Authors:** Shan Huang, Hyun-Chul Jeong

**Affiliations:** 1Department of Physical Education, Jeonbuk National University, Jeonju 54896, Republic of Korea; huangshanjbnu@163.com; 2School of Sport and Health Management, Shaanxi Vocational and Technical College of Finance and Economics, Xianyang 712000, China

**Keywords:** physical education teacher support, self-determined motivation, exercise adherence, college students

## Abstract

Background: Physical exercise is crucial for the physical and mental health of college students, yet improving their exercise adherence remains a pressing challenge. Based on Self-Determination Theory (SDT), physical education (PE) teacher support may enhance students’ self-determined motivation by satisfying their basic psychological needs, thereby promoting exercise adherence. However, the dynamic relationships among teacher support, self-determined motivation, and exercise adherence have not been fully explored. Objective: This study aimed to investigate the dynamic impact of PE teacher support on college students’ exercise adherence and to verify the mediating role of self-determined motivation. Methods: A longitudinal design was adopted, with three waves of data collection (T1, T2, T3) from 555 college students. Cross-lagged models and longitudinal mediation models were constructed to analyze the dynamic relationships among variables. Results: This study found that T1 teacher support significantly and positively predicted T2 self-determined motivation (β = 0.187, *p* < 0.001) and T2 exercise adherence (β = 0.379, *p* < 0.001). Self-determined motivation mediated the relationship between teacher support and exercise adherence (mediation effect = 0.039, 95% CI = [0.007, 0.072]). Additionally, bidirectional predictive relationships existed between self-determined motivation and exercise adherence, although the bidirectional relationship between teacher support and exercise adherence was inconsistent across time points. Conclusion: PE teacher support exerts a significant longitudinal impact on college students’ exercise adherence, primarily through direct effects and the mediating role of self-determined motivation. These findings provide theoretical support for college physical education practices, emphasizing the critical role of teacher support in fulfilling students’ basic psychological needs and enhancing exercise adherence. Future research should expand sample sizes and extend tracking periods to comprehensively reveal the dynamic mechanisms among variables.

## 1. Introduction

In recent years, with the rapid socioeconomic development and improved living standards in China, health issues have become a central societal concern. According to the Healthy China 2030 Plan, the Chinese government explicitly advocates “extensive promotion of national fitness activities and encouragement of physical exercise for key populations”, positioning physical exercise as a vital strategy to enhance public health ([40]). However, despite policy-level efforts to promote fitness, college students’ participation in and adherence to physical exercise remain suboptimal. College students represent a critical population due to their transitional life stage, where declining physical activity coincides with newfound autonomy ([52]). Cross-cultural studies reveal that this trend is particularly pronounced in Asian universities, where adherence rates are 15–20% lower than in Western contexts, underscoring the urgency of culturally tailored interventions. The 2020 National Report on College Students’ Physical Health Monitoring indicates a decline in college students’ physical health, with low exercise adherence being a major factor hindering their physical and mental well-being ([27]). This trend not only impacts individual health outcomes but also poses challenges to the national health strategy.

The Ministry of Education has issued policies such as the Basic Standards for Physical Education in Higher Education Institutions and the Guidelines on Comprehensively Strengthening and Improving School Physical Education in the New Era, emphasizing the need for colleges to enhance physical education curricula, improve PE teachers’ professionalism, and foster lifelong exercise habits through diversified activities and incentives ([26], [28]). Nevertheless, existing studies reveal that despite substantial institutional investments in facilities and curricula, college students’ exercise adherence remains low, primarily due to inadequate psychological support and behavioral guidance ([53]). Thus, improving college students’ exercise adherence through PE teacher support is an urgent priority.

Self-Determination Theory (SDT) provides a robust framework for understanding this issue. SDT posits that individual motivation can be categorized into autonomous and controlled types, with autonomous motivation being a key predictor of behavioral persistence ([45]; [19]). As primary facilitators of college students’ physical exercise, PE teachers can enhance students’ autonomous motivation—and thereby their exercise adherence—through supportive behaviors such as offering autonomy, emotional support, and competence feedback ([18]; [38]; [50]). However, existing research predominantly focuses on static effects of teacher support, leaving the dynamic mechanisms underexplored. Investigating the dynamic impact of PE teacher support on exercise adherence through an SDT lens will deepen theoretical understanding and inform policy optimization. 

Recent SDT interventions in Germany demonstrate that autonomy support (e.g., offering activity choices) increases intrinsic motivation by 31% compared to traditional instruction ([30]). Similarly, [31] ([31]) found relatedness-building group activities were particularly effective in collectivist cultures, suggesting cultural adaptation of teacher strategies.

### 1.1. The Relationship Between PE Teacher Support and College Students’ Exercise Adherence

Self-Determination Theory (SDT) offers a valuable framework for examining the relationship between PE teacher support and exercise adherence. SDT emphasizes the quality—rather than mere intensity—of motivation in goal pursuit ([9]; [22]). In physical education contexts, teacher support significantly influences students’ motivational types and subsequent exercise adherence. PE teachers’ support enhances intrinsic motivation by fulfilling students’ basic psychological needs. According to SDT, these needs include autonomy, competence, and relatedness ([13]; [9]; [34]). When teachers support autonomy through choices, encouragement of self-directed decisions, and positive feedback, students are more likely to experience intrinsic motivation, which correlates with higher exercise adherence due to participation driven by interest and enjoyment ([25]). Teacher support also bolsters exercise adherence by fostering competence. By setting appropriate challenges, providing constructive feedback, and recognizing effort, teachers help students perceive their capabilities and progress. Enhanced competence not only strengthens intrinsic motivation but also promotes identified regulation—where students view exercise as aligned with personal values ([1]; [47])—further supporting long-term adherence.

Additionally, teacher support fulfills relatedness needs by cultivating positive teacher–student relationships and inclusive environments, thereby enhancing students’ sense of belonging and social connection ([3]; [46]). Satisfying relatedness needs improves intrinsic motivation and facilitates the internalization of extrinsic regulation, sustaining exercise behavior ([21]; [50]). Notably, the teacher support–exercise adherence relationship is bidirectional. Students’ adherence may reciprocally influence teacher support ([12]; [23]; [37]). For instance, persistent student engagement may prompt teachers to provide more positive feedback. This dynamic interplay underscores the need for cross-lagged designs to clarify causal relationships. PE teacher support significantly impacts college students’ exercise adherence by fulfilling their basic psychological needs. From the perspective of SDT, this framework allows for a deeper understanding of the mechanisms underlying this relationship and provides guidance for educational practices. However, future research should further explore the dynamic nature of this relationship, along with potential mediating variables and moderating factors, to fully uncover the influence of PE teacher support on college students’ exercise adherence.

### 1.2. The Relationship Between PE Teacher Support and Self-Determined Motivation

When examining the relationship between physical education teacher support and self-determined motivation, Self-Determination Theory (SDT) provides a crucial theoretical framework. SDT emphasizes the quality of individual motivation, conceptualizing it as a continuum ranging from external control to intrinsic self-determination ([10]; [48]). The supportive behaviors of physical education teachers play a pivotal role in fostering students’ self-determined motivation.

Physical education teacher support enhances self-determined motivation by satisfying students’ basic psychological needs. According to SDT, these fundamental needs include autonomy, competence, and relatedness ([13]). When teachers support students’ autonomy by offering choices and encouraging self-directed decision-making, students are more likely to experience intrinsic motivation. This intrinsic motivation is associated with higher participation and adherence, as students engage in physical activities out of genuine interest and enjoyment ([9]; [11]).

Furthermore, teacher support influences self-determined motivation by strengthening students’ sense of competence. Through setting appropriate challenges and providing constructive feedback, teachers help students recognize their capabilities and progress. This enhanced competence not only boosts intrinsic motivation but also facilitates identified regulation—where students perceive exercise as aligned with personal values ([1]). Such identified regulation further reinforces long-term exercise adherence.

Additionally, physical education teacher support plays a vital role in fulfilling students’ relatedness needs. By establishing positive teacher–student relationships and creating supportive, inclusive environments, teachers enhance students’ sense of belonging and social connectedness ([7]; [15]). Satisfying these relatedness needs not only increases intrinsic motivation but also promotes the internalization of extrinsic regulation, thereby sustaining exercise behavior ([17]; [43]).

### 1.3. The Relationship Between Self-Determined Motivation and College Students’ Exercise Adherence

According to SDT, the quality of motivation rather than mere motivational intensity serves as a stronger predictor of behavioral persistence ([10]; [48]). SDT categorizes motivation into intrinsic motivation, extrinsic motivation (including identified regulation, introjected regulation, and external regulation), and amotivation, forming a continuum from high to low self-determination ([13]). In college students’ exercise behaviors, the level of self-determined motivation directly influences their ability to maintain long-term exercise adherence.

First, intrinsic motivation represents one of the strongest predictors of exercise adherence. Intrinsic motivation refers to engaging in activities due to inherent interest, enjoyment, or satisfaction ([6]). Research indicates that college students with higher intrinsic motivation are more likely to view exercise as an enjoyable activity rather than a result of external pressure, thus more readily developing long-term exercise habits ([44]). For instance, a study on college students found that intrinsic motivation significantly predicted their exercise frequency and duration. This suggests that when students experience pleasure and a sense of achievement during exercise, they demonstrate greater persistence.

Second, identified regulation within extrinsic motivation also plays a crucial role in exercise adherence. Identified regulation occurs when individuals perceive a behavior as aligned with their personal values or goals, even if the activity itself may not be entirely pleasurable. Among college students, this typically manifests as viewing exercise as part of a healthy lifestyle or a means to achieve personal objectives (e.g., weight loss or improved fitness) ([4]). Studies show that identified regulation significantly correlates with long-term exercise maintenance because this motivational type possesses relatively high self-determination, enabling individuals to overcome short-term difficulties or burnout ([20]).

In contrast, motivations with low self-determination (such as external and introjected regulation) and amotivation typically correlate with poorer exercise adherence. External regulation refers to engaging in activities due to external rewards or punishments, while introjected regulation stems from internal pressures (e.g., guilt or anxiety) ([6]). Although these motivational types may prompt college students to exercise in the short term, they often fail to sustain long-term exercise behaviors due to lacking intrinsic drive ([29]). Amotivation, characterized by the absence of clear goals or willingness to exercise, shows a strong association with exercise discontinuation ([6]).

In previous academic discussions, researchers have primarily focused on static effects, while dynamic longitudinal relationships have not received adequate attention or thorough analysis. Therefore, this study adopts a cross-lagged model to deepen the analysis of temporal dynamics, exploring the interplay and impact between teacher support and student motivation over time.

## 2. Methods

### 2.1. Participants

This study adopted a longitudinal design to dynamically examine college students’ exercise adherence and its influencing factors across three time points (T1, T2, T3). The participants were full-time undergraduate students from a university, selected through stratified random sampling to ensure sample representativeness. The initial measurement (T1) was conducted in September 2022, with 600 questionnaires distributed and 580 valid responses collected. The second measurement (T2) was conducted in January 2023, yielding 565 valid responses, and the third measurement (T3) was conducted in May 2023, with 555 valid responses obtained. The final analytical sample consisted of 555 participants, including 278 males (50.1%) and 277 females (49.9%), aged 18 to 24 years (mean = 20.3, SD = 1.2). The participants’ majors covered STEM disciplines, humanities, and social sciences, ensuring sample diversity. To mitigate the impact of sample attrition on the results, multiple imputation was used to handle missing data, and cross-lagged modeling was employed to analyze the dynamic relationship between PE teacher support and college students’ exercise adherence. This study was approved by the university’s ethics committee, and all participants provided informed consent, with data anonymity and confidentiality guaranteed. The research protocol was approved by the Ethics Committee of Jeonbuk National University (2024-12-31-J-E002398) and adhered to the ethical standards outlined in the 1964 Declaration of Helsinki and its later amendments. All participants signed an informed consent form.

### 2.2. Measures

Perceived Autonomy Support Scale for PE Teachers: This study used a self-developed Perceived Autonomy Support Scale for PE Teachers to measure teacher support. The scale consisted of six items, such as “I feel my PE teacher understands me in sports” and “My PE teacher gives me confidence in sports, making me believe I can do well.” A 5-point Likert scale was used, ranging from 1 (“strongly disagree”) to 5 (“strongly agree”). Autonomy support refers to the degree to which PE teachers provide students with a sense of autonomy during physical education activities. This scale was used to measure students’ perceived autonomy support from PE teachers in class. The scale demonstrated good reliability and validity among Chinese college students ([8]). In this study, the scale’s reliability was 0.955.

Sport Motivation Scale: This study employed Fang Wenxuan’s adapted version of Su Yu’s Sport Motivation Scale to measure college students’ self-determined motivation ([8]; [41]; [42]). Fang modified the content to suit school sports settings. The scale consisted of 18 items, including five subscales: intrinsic motivation (three items, e.g., “I find sports participation enjoyable”), identified regulation (four items, e.g., “Because it brings me happiness”), introjected regulation (four items, e.g., “Because quitting would make me feel guilty”), external regulation (four items, e.g., “I have to participate in sports for some reason”), and amotivation (three items, e.g., “I participate in sports, but I don’t know if it’s worth doing”). A 5-point Likert scale was used, ranging from 1 (“strongly disagree”) to 5 (“strongly agree”). The scale demonstrated good reliability and validity among Chinese college students ([8]). The reliability of this scale in this study was 0.918.

Exercise Adherence Questionnaire: This study used the Exercise Adherence Questionnaire developed by ([51]). Through interviews and open-ended surveys. The questionnaire consisted of three dimensions—behavioral habits, effort investment, and emotional experience—with a total of 14 items. A 5-point Likert scale was used, ranging from 1 (“strongly disagree”) to 5 (“strongly agree”). The scoring method involved summing the item scores for each dimension and the total scale, then calculating the mean. Higher mean scores indicated higher levels of exercise adherence. The scale demonstrated good reliability and validity among Chinese college students ([36]; [51]). The reliability coefficient of this scale in the current study was 0.974.

### 2.3. Data Analysis

This study used SPSS 26.0 for descriptive statistics, correlation analysis, and common method bias testing, and AMOS 24.0 for cross-lagged analysis.

## 3. Results

### 3.1. Common Method Bias Test

The Harman single-factor test was conducted to evaluate potential common method bias. Results revealed that across the three measurements, there were four, five, and five factors with eigenvalues exceeding 1, respectively. The variance explained by the first factor was 27.97%, 30.40%, and 35.52% for each measurement, all below the critical threshold of 40%. These findings indicate no substantial common method bias in the three rounds of data collection.

### 3.2. Descriptive Statistics and Correlation Analysis of Variables

Table 1 presents the descriptive statistics and correlation analysis of variables across three time points (T1, T2, T3). The sample size was 555. The means (M) and standard errors (SE) indicate fluctuations in teacher support, self-determined motivation, and exercise adherence across time points. Correlation analysis revealed significant positive associations (*p* < 0.01) both within and between time points. Examples are presented below:

Teacher Support T1 correlated significantly with Self-Determined Motivation T1 (r = 0.513) and Exercise Adherence T1 (r = 0.564).

Teacher Support T2 correlated significantly with Self-Determined Motivation T2 (r = 0.486) and Exercise Adherence T2 (r = 0.531).

Teacher Support T3 correlated significantly with Self-Determined Motivation T3 (r = 0.530) and Exercise Adherence T3 (r = 0.593).

Cross-temporal correlations also showed significant associations, such as between Teacher Support T1 and Exercise Adherence T3 (r = 0.589). These results provide foundational support for subsequent cross-lagged analyses.

### 3.3. Cross-Lagged Analysis of Physical Education Teacher Support, Self-Determined Motivation, and College Students’ Exercise Adherence

To explore the reciprocal relationships among physical education teacher support, self-determined motivation, and college students’ exercise adherence, a cross-lagged model was constructed (Figure 1). The model demonstrated acceptable fit: χ^2^/df = 7.590, *p* < 0.001, CFI = 0.922, GFI = 0.947, TLI = 0.889, SRMR = 0.077, RMSEA = 0.173.

The results in Figure 1 show that T1 teacher support significantly and positively predicted T2 self-determination motivation (β = 0.187, *p* < 0.001); T1 teacher support significantly and positively predicted T2 exercise adherence (β = 0.379, *p* < 0.001); T1 self-determination motivation significantly and positively predicted both T2 teacher support (β = 0.094, *p* < 0.05) and T2 exercise adherence (β = 0.211, *p* < 0.001); and T1 exercise adherence significantly and positively predicted both T2 teacher support (β = 0.188, *p* < 0.001) and T2 self-determination motivation (β = 0.286, *p* < 0.001).

Additionally, T2 teacher support had no significant effect on either T3 self-determination motivation (β = 0.084, *p* > 0.05) or T3 exercise adherence (β = 0.041, *p* > 0.05); T2 self-determination motivation significantly and positively predicted both T3 teacher support (β = 0.182, *p* < 0.001) and T3 exercise adherence (β = 0.332, *p* < 0.001); and T2 exercise adherence significantly and positively predicted both T3 teacher support (β = 0.308, *p* < 0.001) and T3 self-determination motivation (β = 0.130, *p* < 0.01).

The results demonstrate significant bidirectional predictions between self-determination motivation and exercise adherence, while the bidirectional predictions between teacher support and self-determination motivation/exercise adherence were inconsistent.

### 3.4. Longitudinal Mediating Effect of Self-Determined Motivation

To further examine the interrelationships among teacher support, self-determined motivation, and exercise adherence, this study constructed a longitudinal mediation model for verification, as shown in Figure 2.

Bootstrap test results (Table 2) further verified the significance of the mediation effects.

The total effect was 0.589 (95% CI = [0.521, 0.656]), and the direct effect was 0.502 (95% CI = [0.429, 0.575]), indicating that the direct effect of teacher support on exercise adherence was supported.

The mediation effect of teacher support T1 on exercise adherence T3 through self-determined motivation T2 was 0.039 (95% CI = [0.007, 0.072]). The confidence intervals of the mediation effects did not include 0, demonstrating that the mediation effect was statistically significant.

These results suggest that teacher support can not only directly influence individuals’ exercise adherence but also indirectly facilitate the development of exercise adherence by enhancing self-determined motivation.

## 4. Discussion

This study, grounded in Self-Determination Theory (SDT), employed cross-lagged and longitudinal mediation models to thoroughly examine the longitudinal effects of physical education teacher support on college students’ exercise adherence. The findings demonstrate that teacher support not only directly facilitates students’ short-term exercise adherence but also exerts long-term indirect effects by enhancing self-determined motivation. Below, we analyze these mechanisms through the lens of SDT, integrating our results with existing research.

### 4.1. Theoretical Framework of Self-Determination Theory

Proposed by [6] ([6]), SDT classifies human motivation into intrinsic motivation, extrinsic motivation, and amotivation, with intrinsic motivation serving as the core driver of behavioral persistence ([6]). The theory posits that satisfaction of basic psychological needs (autonomy, competence, and relatedness) is crucial for developing and maintaining intrinsic motivation. When external environments (e.g., teacher support) fulfill these needs, individuals’ intrinsic motivation and behavioral persistence significantly strengthen ([33]). In physical activity contexts, SDT has been widely applied to explain the formation of exercise motivation and adherence ([35]; [44]).

### 4.2. Immediate Effects of Teacher Support: Satisfying Basic Psychological Needs

Our results show that T1 teacher support significantly predicted T2 exercise adherence, indicating direct motivational effects in the short term. This aligns with SDT’s proposition that external support (e.g., from teachers) directly enhances behavioral persistence by satisfying basic psychological needs ([33]). Autonomy Need: Teachers foster autonomy by encouraging students to choose activities, set plans, and participate in decision-making, thereby enhancing motivation ([14]). For instance, offering diverse exercise options and respecting preferences boosts intrinsic motivation ([16]; [21]). Competence Need: Personalized guidance, appropriately challenging tasks, and positive feedback strengthen competence and adherence ([35]; [44]). Tailoring tasks to skill levels and providing encouragement are key ([5]; [49]). Relatedness Need: Positive classroom climates, strong teacher–student relationships, and peer interactions fulfill relatedness needs ([16]). Group activities enhance belongingness and motivation ([32]).

### 4.3. Mediating Role of Self-Determined Motivation: A Long-Term Mechanism

T1 teacher support indirectly influenced T3 exercise adherence through T2 self-determined motivation, confirming SDT’s basic psychological needs theory ([6]). External support satisfies autonomy, competence, and relatedness, thereby strengthening intrinsic motivation. For example, autonomy-supportive teaching and constructive feedback enhance self-determined motivation, promoting adherence ([14]) Bootstrap tests validated this longitudinal mediation (effect = 0.039, 95% CI [0.007, 0.072]), consistent with [39] ([39]). Emotional support and individualized instruction elevate self-determined motivation, sustaining adherence ([16]; [21]).

### 4.4. Dynamics of Longitudinal Effects: Temporal and Contextual Variability

While T2 teacher support did not significantly predict T3 outcomes, T2 self-determined motivation and adherence influenced T3 teacher support and adherence. This instability may reflect changing student needs or environmental factors ([5]; [49]). Teachers must adopt support strategies to sustain adherence. Prior studies corroborate these longitudinal effects. [39] ([39]) found teacher support enhanced adherence via self-determined motivation ([2]; [24]), aligning with our results.

### 4.5. Limitations

Although this study provides important insights into the dynamic impact of physical education teacher support on college students’ exercise adherence, there are still some limitations. First, the sample was drawn from only one university, and the relatively limited sample size may restrict the generalizability of the findings. Future research should expand the sample scope to include universities from different regions and of different types to enhance the representativeness and applicability of the results. 

Second, this study employed a longitudinal tracking design, which, while revealing dynamic relationships between variables to some extent, may not fully capture long-term trends due to the short time intervals. Future studies could extend the tracking period and increase the number of measurement points to more comprehensively uncover long-term relationships among variables. 

Additionally, this study focused solely on the relationships between physical education teacher support, self-determined motivation, and exercise adherence, without considering other potential influencing factors such as peer support, family support, and individual characteristics. Future research could further explore the roles of these factors in the relationship between teacher support and exercise adherence to gain a more comprehensive understanding of the dynamic mechanisms underlying college students’ exercise adherence. 

Finally, this study primarily relied on self-reported data, which may be subject to social desirability bias.

## 5. Conclusions

This study was based on Self-Determination Theory and employed cross-lagged and longitudinal mediation models to explore the dynamic impact of physical education teacher support on college students’ exercise adherence. The findings indicate that teacher support not only directly enhances students’ exercise adherence but also indirectly improves it by strengthening self-determined motivation. This result validates the mediating role of self-determined motivation between teacher support and exercise adherence while also revealing the dynamic mechanism through which teacher support influences exercise adherence at different time points.

## Figures and Tables

**Figure 1 behavsci-15-00802-f001:**
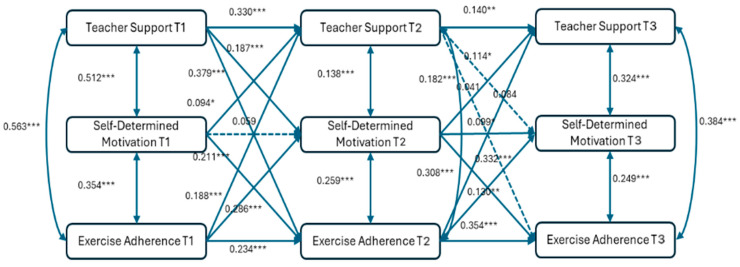
Cross-lagged analysis of physical education teacher support, self-determination motivation, and college students’ exercise adherence. * *p* < 0.05; ** *p* < 0.01; *** *p* < 0.001.

**Figure 2 behavsci-15-00802-f002:**
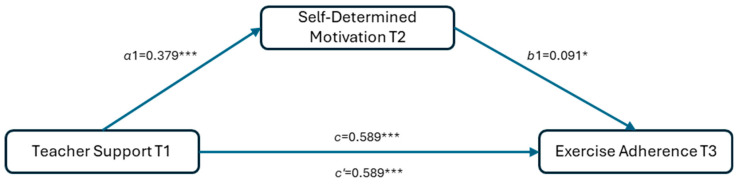
Longitudinal mediation model of self-determined motivation. * *p* < 0.05; *** *p* < 0.001.

**Table 1 behavsci-15-00802-t001:** Descriptive statistics and correlation analysis of variables.

Variable	N	M	SE	1	2	3	4	5	6	7	8	9
Teacher support T1	555	38.09	11.356	1								
Self-Determination motivation T1	555	30.74	7.056	0.513 **	1							
Exercise adherence T1	555	43.86	7.746	0.546 **	0.354 **	1						
Teacher support T2	555	36.03	10.341	0.484 **	0.330 **	0.407 **	1					
Self-Determination motivation T2	555	30.78	6.707	0.379 **	0.257 **	0.413 **	0.486 **	1				
Exercise adherence T2	555	38.68	12.128	0.619 **	0.488 **	0.523 **	0.531 **	0.433 **	1			
Teacher support T3	555	35.46	11.373	0.514 **	0.299 **	0.412 **	0.359 **	0.315 **	0.432 **	1		
Self-Determination motivation T3	555	32.13	7.623	0.480 **	0.337 **	0.325 **	0.308 **	0.283 **	0.419 **	0.530 **	1	
Exercise adherence T3	555	42.07	7.826	0.589 **	0.343 **	0.439 **	0.292 **	0.303 **	0.432 **	0.593 **	0.448 **	1

Note: ** *p* < 0.001; T1, T2, and T3 denote Time 1, Time 2, and Time 3, respectively.

**Table 2 behavsci-15-00802-t002:** Bootstrap test for the longitudinal mediating effect of self-determined motivation.

		95% CI			
Effect type	Effect size	Lower	Upper	SE	Conclusion
Total effect	0.589	0.521	0.656	0.034	
Direct effect	0.502	0.429	0.575	0.037	Supported
Teacher support T1→Self-Determination motivation T2→Exercise adherence T3	0.039	0.007	0.072	0.016	Supported

## Data Availability

The data that support the findings of this study are available from the corresponding author H.-C.J. upon reasonable request.

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
