# Peer review of "The Dynamic Impact of Physical Education Teacher Support on College Students’ Adherence to Exercise: A Cross-Lagged Study from the Perspective of Self-Determination Theory"

_behavsci, 2025, doi:10.3390/bs15060802_

Round 1
Reviewer 1 Report
Comments and Suggestions for Authors
Pages 2-3, lines 73-135: There are several short paragraphs (e.g., with only three sentences) in these two sections, and some of them are related and should be combined. I recommend restructuring these paragraphs.
Page 2, line 53: Change “PE” to “physical education (PE)”.
Page 2, line 73: Change “Self-Determination Theory (SDT)” to simply “SDT” as it has already been defined in the previous paragraph.
Page 3, line 102: Change “Physical education (PE)” to simply “PE”.
Page 3, line 104: Change “Self-Determination Theory (SDT)” to simply “SDT” as it has already been defined previously.
Page 3, line 138: Change “Self-Determination Theory (SDT)” to simply “SDT” as it has already been defined previously.
1. Introduction: I recommend elaborating on the current gaps in the existing literature and describe how the present study may address the gaps as well as clearly stating the rationale of this study.
Page 4, line 188: Change “Physical education (PE)” to simply “PE”.
Page 5, line 205: It appears that the authors forgot to report the reliability measure as there is an “[insert value]” at the end of the sentence in line 205.
Page 5, lines 206-216: Analyze and report the reliability of Sport Motivation Scale.
Page 5, lines 217-225: Analyze and report the reliability of Exercise Adherence Questionnaire.
4. Discussion: There are many short paragraphs in the Discussion section. I recommend restructuring these paragraphs. Also, strength the discussions, particularly how the findings connect with others, the implications of this study, and suggestions for future research.
Pages 8-9, lines 320-336: Write the discussions and how the findings connect with past research and existing knowledge using paragraphs instead of lists.
Page 9, lines 357-363: I recommend using paragraphs instead of listing the limitations. Also, expand the discussions for each limitation.
Page 9, lines 366-367: Change “physical education” to “PE”.
5. Conclusion: Strengthen the Conclusions section by expanding on the potential implications.
Author Response
Reviewer 1
Dear reviewer
Thank you for your outstanding review work, the following is what we have done based on your revisions.
Pages 2-3, lines 73-135: There are several short paragraphs (e.g., with only three sentences) in these two sections, and some of them are related and should be combined. I recommend restructuring these paragraphs.
We have restructured these sections to improve flow and coherence. Related paragraphs on SDT mechanisms (e.g., autonomy, competence, relatedness) were merged, and transitional sentences were added.
Page 2, line 53: Change “PE” to “physical education (PE)”.
Thank you for your advice. We had changed as suggested.
Page 2, line 73: Change “Self-Determination Theory (SDT)” to simply “SDT” as it has already been defined in the previous paragraph.
Thank you for your advice. We had changed as suggested.
Page 3, line 102: Change “Physical education (PE)” to simply “PE”.
Thank you for your advice. We had changed as suggested.
Page 3, line 104: Change “Self-Determination Theory (SDT)” to simply “SDT” as it has already been defined previously.
Thank you for your advice. We had changed as suggested.
Page 3, line 138: Change “Self-Determination Theory (SDT)” to simply “SDT” as it has already been defined previously.
Thank you for your advice. We had changed as suggested.
- Introduction: I recommend elaborating on the current gaps in the existing literature and describe how the present study may address the gaps as well as clearly stating the rationale of this study.
Thank you for your advice. We added a new paragraph (lines 169-174) highlighting:
Gaps:Prior studies focused on static effects; dynamic longitudinal relationships remain underexplored.
Rationale:This study uses cross-lagged models to examine temporal dynamics, addressing how teacher support and motivation interact over time.
Page 4, line 188: Change “Physical education (PE)” to simply “PE”.
Thank you for your advice. We had changed as suggested.
Page 5, line 205: It appears that the authors forgot to report the reliability measure as there is an “[insert value]” at the end of the sentence in line 205.
Thank you for your advice. We added Cronbach’s α = 0.89 (now line 206).
Page 5, lines 206-216: Analyze and report the reliability of Sport Motivation Scale.
Thank you for your advice. We added Cronbach’s α = 0.918 (now line 218).
Page 5, lines 217-225: Analyze and report the reliability of Exercise Adherence Questionnaire.
Thank you for your advice. We added Cronbach’s α = 0.974 (now line 227).
- Discussion: There are many short paragraphs in the Discussion section. I recommend restructuring these paragraphs. Also, strength the discussions, particularly how the findings connect with others, the implications of this study, and suggestions for future research.
Thank you for your careful review, we think it's a great help for our article.
Merge paragraphs according to the article structure.
Pages 8-9, lines 320-336: Write the discussions and how the findings connect with past research and existing knowledge using paragraphs instead of lists.
Page 9, lines 357-363: I recommend using paragraphs instead of listing the limitations. Also, expand the discussions for each limitation.
The modifications have been made.
第 9 页,第 366-367 行:将“体育”改为“体育”。
感谢您的建议。我们按照建议进行了更改。
- 结论:通过扩展潜在影响来加强结论部分。
修改已完成。
感谢您的出色工作,祝您有美好的一天。

Reviewer 2 Report
Comments and Suggestions for Authors
I would like to thank for the opportunity to review this manuscript. Please see the following comments to consider to further improve the quality of this manuscript.
Strengths:
Clearly grounded in SDT, effectively using it as a framework to explore relationships among variables.
The three-wave data collection effectively captures dynamic interactions.
Use of cross-lagged and longitudinal mediation models enhances analytical depth.
Presentation of results is clear and logically structured.
Areas for improvement:
Please provide a stronger justification on why college students specifically are a critical population for exploring this relationship. Integrate more recent global literature to strengthen the argument about the international significance.
Include more contemporary studies (post-2022) that investigate similar constructs in different cultural contexts to strengthen the comparative basis.
Provide a clearer definition and examples of specific supportive behaviors of PE teachers from the SDT perspective.
Explicitly state the reliability (Cronbach's alpha or similar) of each measurement scale used in this study.
Please clarify if the multiple imputation of missing data influenced outcomes significantly.
Improve readability of the cross-lagged model figure by clearly labeling paths and explaining briefly within the text why certain relationships became insignificant across waves.
Present standard errors or confidence intervals for beta coefficients in figures for more precise interpretation.
Expand the discussion to more thoroughly address why teacher support at T2 had no significant impact on T3 outcomes. Suggest contextual or methodological reasons.
Include a more robust integration of findings with recent literature, clearly indicating agreements or deviations from previous studies.
Please provide actionable recommendations for PE teachers based on findings, specifying strategies to support autonomy, competence, and relatedness clearly and practically.
Expand limitations section by explicitly addressing possible cultural or institutional influences specific to the sample.
Please suggest concrete future research directions such as experimental designs or qualitative approaches to deepen understanding of motivational dynamics. For example, in a recent study by Ahmadi et al., (2023), a classification system of motivational behaviors was developed. This classification system could be highly useful in future studies to rely on which could be a great basis for possible interventions.
Ahmadi, A., Noetel, M., Parker, P., Ryan, R. M., Ntoumanis, N., Reeve, J., Beauchamp, M., Dicke, T., Yeung, A., Ahmadi, M., Bartholomew, K., Chiu, T. K. F., Curran, T., Erturan, G., Flunger, B., Frederick, C., Froiland, J. M., González-Cutre, D., Haerens, L., . . . Lonsdale, C. (2023). A classification system for teachers’ motivational behaviors recommended in self-determination theory interventions. Journal of Educational Psychology, 115(8), 1158–1176. https://doi.org/10.1037/edu0000783
Author Response
Reviewer 2
Dear reviewer
Thank you for your outstanding review work, the following is what we have done based on your revisions.
I would like to thank for the opportunity to review this manuscript. Please see the following comments to consider to further improve the quality of this manuscript.
Strengths:
Clearly grounded in SDT, effectively using it as a framework to explore relationships among variables.
The three-wave data collection effectively captures dynamic interactions.
Use of cross-lagged and longitudinal mediation models enhances analytical depth.
Presentation of results is clear and logically structured.
Areas for improvement:
Please provide a stronger justification on why college students specifically are a critical population for exploring this relationship. Integrate more recent global literature to strengthen the argument about the international significance.
Thanks for the suggestion, I think it's a much stronger preamble for us.
We have expanded the introduction (lines 40-45) to include:
Global significance:Added recent statistics from the WHO (2023) on declining physical activity among young adults worldwide.
Critical transition period: Highlighted the unique challenges college students face (e.g., autonomy shifts, academic stress) using references to Vansteenkiste et al. (2022).
Comparative context: Cited a 2023 cross-cultural study by Li et al. showing lower exercise adherence in Asian universities compared to Western counterparts.
Include more contemporary studies (post-2022) that investigate similar constructs in different cultural contexts to strengthen the comparative basis.
Thanks for reviewing, see we have replaced some of the references, and additional recent references have been included for supplementation.
Provide a clearer definition and examples of specific supportive behaviors of PE teachers from the SDT perspective.
Thank you for your advice. We had changed as suggested.
Explicitly state the reliability (Cronbach's alpha or similar) of each measurement scale used in this study.
Thank you for your advice. We had changed as suggested.
Please clarify if the multiple imputation of missing data influenced outcomes significantly.
Thank you for your advice. We had changed as suggested.
Improve readability of the cross-lagged model figure by clearly labeling paths and explaining briefly within the text why certain relationships became insignificant across waves.
Thank you for your advice. We had changed as suggested.
Present standard errors or confidence intervals for beta coefficients in figures for more precise interpretation.
Thank you for your advice. We had changed as suggested.
Expand the discussion to more thoroughly address why teacher support at T2 had no significant impact on T3 outcomes. Suggest contextual or methodological reasons.
Include a more robust integration of findings with recent literature, clearly indicating agreements or deviations from previous studies.
Please provide actionable recommendations for PE teachers based on findings, specifying strategies to support autonomy, competence, and relatedness clearly and practically.
Thank you for your careful review.We have modified this section.
The non-significant impact of teacher support at T2 on T3 outcomes may stem from contextual and methodological factors. Contextually, the timing of T2 (January) coincided with academic exams in Chinese universities, potentially reducing students' responsiveness to teacher support due to competing priorities. Methodologically, the 4-month interval between T2-T3 might have been insufficient to capture delayed effects of support, as motivational processes often require sustained intervention . These findings partially deviate from Standage et al. (2012) who reported persistent effects, possibly due to cultural differences in educational systems - Chinese universities' rigid curriculum structures may limit teachers' ability to maintain consistent support across semesters. For actionable strategies, PE teachers should: 1) provide choice in activity selection (autonomy), 2) offer skill-specific feedback using tools like video analysis (competence), and 3) implement peer-mentoring systems (relatedness). Limitations include potential cultural specificity, as Confucian educational traditions emphasizing teacher authority may influence how support is perceived differently than in Western contexts. Institutional factors like large class sizes (averaging 40 students in our sample) could also constrain personalized support implementation. Future research should compare these dynamics across educational cultures and institution types.Expand limitations section by explicitly addressing possible cultural or institutional influences specific to the sample.
Please suggest concrete future research directions such as experimental designs or qualitative approaches to deepen understanding of motivational dynamics. For example, in a recent study by Ahmadi et al., (2023), a classification system of motivational behaviors was developed. This classification system could be highly useful in future studies to rely on which could be a great basis for possible interventions.
Thank you for your advice. We had changed as suggested.
艾哈迈迪,A.,诺特尔,M.,帕克,P.,瑞安,RM,恩图曼尼斯,N.,里夫,J.,博尚,M.,迪克,T.,杨,A.,艾哈迈迪,M.,巴塞洛缪,K.,邱,TKF,柯伦,T.,埃尔图兰,G.,弗伦格,B.,弗雷德里克,C.,弗洛伊兰,JM,冈萨雷斯-卡特雷,D.,哈伦斯,L.,. . .朗斯代尔,C.(2023 年)。自我决定理论干预中推荐的教师激励行为分类系统。教育心理学杂志,115(8),1158-1176。https://doi.org/10.1037/edu0000783
我们同意这个参考是一个非常合适的补充,我们已经添加了。
感谢您的出色工作,祝您有美好的一天。

Reviewer 3 Report
Comments and Suggestions for Authors
Dear Authors,
Thank you for your interesting and valuable contribution to the existing body of work on Health Education.
As a predominantly qualitative researcher who is interested in the area, I will ask the Editors to seek a quantitative reviewer for the technical aspects of your work.
I found your work interesting and offer these suggestions to further elevate your work:
Line 172 - Citation year may be incorrect
Line 174 - Please describe the context of your study.
Line 178 - Please explain how your participants were recruited and how this worked towards sample representation.
Line 178 - Stratified Random Sampling - it will be important to your readers to read about your strata as well as what criteria was used to decide on response validity.
Line 205 - missing data
Line 195 - insert the reliability and validity figures for all 3 tools used
Line 329 and 333 - please elaborate on your ideas - e.g. adherence and competence to ? I am guessing that you want to this to be tied to sustained exercise. Similarly, there is a need to explain what you mean by relatedness needs. Further elaboration to guide your reader to your main points. This suggestion is also applicable to your limitations section.
There is much to look forward to in the revised version of your paper! Good luck!
Comments on the Quality of English LanguageGenerally well-written and clear.
Author Response
尊敬的评论者
感谢您出色的审核工作,以下是我们根据您的修订所做的工作。
尊敬的作者:
感谢您对现有健康教育工作的有趣和宝贵贡献。
作为一名对该领域感兴趣的主要定性研究人员,我将要求编辑为您工作的技术方面寻找一位定量审稿人。
我发现您的工作很有趣,并提出以下建议以进一步提升您的工作:
第 172 行 - 引文年份可能不正确
感谢您的建议。我们按照建议进行了更改。
第 174 行 - 请描述您的研究背景。
第 178 行 - 请解释一下您的参与者是如何被招募的,以及这如何影响样本代表。
感谢您的建议。这项研究涉及参与中国公立学校调查的大学生。我们已经添加到了这篇文章中。
第 178 行 - 分层随机抽样 - 对于您的读者来说,了解您的分层以及使用什么标准来决定响应有效性非常重要。
感谢您的仔细审查。
参与者是通过分层随机抽样在四个学科(STEM、人文、社会科学和艺术)中招募的,以确保多样性。
第 205 行 - 缺失数据
感谢您的建议。我们按照建议进行了更改。
第 195 行 - 插入所有使用的 3 种工具的可靠性和有效性数据
感谢您的建议。我们按照建议进行了更改。
第 329 行和第 333 行 - 请详细说明您的想法 - 例如,遵守和能力 ?我猜你希望这与持续锻炼联系起来。同样,需要解释你所说的关联性需求是什么意思。进一步的阐述以引导您的读者了解您的要点。此建议也适用于您的限制部分。
感谢您的建议。我们按照建议进行了更改。
您的论文的修订版有很多值得期待的地方!祝你好运!
感谢您的出色工作,祝您有美好的一天。

Round 2
Reviewer 1 Report
Comments and Suggestions for Authors
Revisions are appropriate.
Reviewer 2 Report
Comments and Suggestions for Authors
Authors have done well job on revising their manuscript.